# Lysyl Oxidase-like 2-Guided Benefits of Early Cardiac Rehabilitation in Acute Heart Failure: A Prospective Cohort Study in Taiwan

**DOI:** 10.3390/biomedicines13092228

**Published:** 2025-09-10

**Authors:** Shyh-Ming Chen, Lin-Yi Wang, Hao-Yi Hsiao, Chin-Ling Wei, You-Cheng Zheng, Po-Jui Wu, Chien-Jen Chen, Chi-Ling Hang, Steve Leu, Yung-Lung Chen

**Affiliations:** 1Section of Cardiology, Department of Internal Medicine, Heart Failure Center, Kaohsiung Chang Gung Memorial Hospital, Kaohsiung 83301, Taiwan; 2College of Medicine, Chang Gung University, Taoyuan 33302, Taiwan; 3Department of Physical Medicine and Rehabilitation, Kaohsiung Chang Gung Memorial Hospital, Kaohsiung 83301, Taiwan; 4Department of Nursing, Heart Failure Center, Kaohsiung Chang Gung Memorial Hospital, Kaohsiung 83301, Taiwan; 5Department of Biotechnology, Kaohsiung Medical University, Kaohsiung 80708, Taiwan; 6Institute for Translational Research in Biomedicine, Kaohsiung Chang Gung Memorial Hospital, Kaohsiung 83301, Taiwan

**Keywords:** acute heart failure, cardiac rehabilitation, lysyl oxidase-like 2, mortality

## Abstract

**Background/Objectives:** The objectives of this study were to determine whether the early initiation of cardiac rehabilitation (CR) within 6 weeks of discharge improves long-term outcomes in patients hospitalized for acute heart failure (HF) and to evaluate whether baseline lysyl oxidase-like 2 (LOXL2) levels affect the response to CR. **Methods:** We prospectively enrolled patients with acute HF who completed a structured Heart Failure Disease Management Program between January 2019 and July 2022. Participants were categorized into an early-CR group (initiating supervised CR within 6 weeks post-discharge and continuing a home-based program) or a non-CR group. The primary outcome was all-cause mortality. The secondary outcomes included HF rehospitalization and changes in scores on the 12-item Kansas City Cardiomyopathy Questionnaire (KCCQ-12) at 6 and 12 months. A post hoc analysis was conducted to stratify patients by baseline LOXL2 levels in order to assess differential CR effects in relation to the severity of cardiac fibrosis. **Results:** Out of 162 patients, 34 participated in early CR. After 1:1 propensity score matching, each group contained 33 patients. Over a median follow-up of 2.85 years, the early-CR group experienced lower all-cause mortality (0 vs. 87.2 events per 1000 patient-years; rate difference: −0.087). A subgroup analysis revealed the greatest benefit among patients with LOXL2 levels > 200 pg/mL (0 vs. 172.3 events per 1000 patient-years; rate difference: −0.172). **Conclusions:** Early post-discharge CR was associated with improved survival in patients with acute HF. The survival benefit was more pronounced in patients with an elevated level of LOXL2, suggesting its potential role as a biomarker for fibrosis-guided CR strategies. Health systems seeking scalability may consider embedding exercise-based and biomarker-guided CR programs within clinical networks early on to improve access while advancing patient-centered care. Further randomized trials are warranted to confirm these findings.

## 1. Introduction

Heart Failure (HF) affects an estimated 64 million people worldwide. While the incidence has plateaued in some high-income settings, the prevalence continues to rise due to an aging population and improved survival rates following acute cardiovascular events [1]. A study using the Taiwan National Health Insurance Research Database indicated an increase in the prevalence of HF from 0.63% in 2001 to 1.40% in 2016, representing a 2.22-fold increase over 16 years [2]. Despite medical advancements, the clinical prognosis of HF has not improved proportionately, with the estimated mortality rates of newly diagnosed HF being 38.5% at 2 years and 75.5% at 10 years [2]. Thus, reducing mortality in acute HF and preventing rehospitalization for HF are critical challenges in modern medicine.

Cardiac rehabilitation (CR) is widely recognized as an essential tool for the secondary prevention of HF. CR programs usually include organized exercise training, which has been consistently proven to improve exercise capacity in individuals with HF. Improved exercise tolerance is linked to better overall cardiovascular function, reduced symptoms, and enhanced quality of life [3]. The “Heart Failure: A Controlled Trial Investigating Outcomes of Exercise Training” (HF-ACTION) trial found that aerobic exercise training improved the quality of life of patients with HF with a reduced ejection fraction (HFrEF), and, after prespecified adjustment, exercise training appeared to be associated with reductions in cardiovascular mortality or hospitalization for HF [4]. The HF-ACTION trial provided evidence supporting CR for stable chronic HF in patients with HFrEF categorized as New York Heart Association (NYHA) classes II–IV despite being on optimal therapy for ≥6 weeks. A retrospective study also suggested that early CR (within <6 weeks of discharge) may have beneficial effects on patients with acute HF who have just been discharged [5]. The “Rehabilitation Therapy in Older Acute Heart Failure Patients” (REHAB-HF) trial is a multicenter, single-blind, randomized controlled trial (RCT) that assessed the effect of exercise training in older patients hospitalized for acute HF. In the REHAB-HF trial, intervention was initiated during or soon after hospitalization and continued for phase II CR after discharge. The primary endpoint was scores on the Short Physical Performance Battery (SPPB) at 3 months. The results showed least-squares mean (±standard error) SPPB scores of 8.3 ± 0.2 in the intervention group and 6.9 ± 0.2 in the control group at 3 months (mean between-group difference: 1.5; 95% confidence interval [CI]: 0.9–2.0; *p* < 0.001) [6]. Current evidence remains insufficient to definitively assess the impact of early rehabilitation on clinical hard endpoints in patients with acute HF, and the long-term effects of early rehabilitation in patients with acute HF have not been fully investigated.

The presence and extent of cardiac fibrosis substantially affect the prognosis of HF, as it can lead to contractile dysfunction and arrhythmic events [7]. Cardiac fibrosis is considered a key therapeutic target in HF management [8]. Nevertheless, existing therapies aiming to reverse cardiac fibrosis generally focus on nonspecific aspects of the fibrotic process, and the resilience of fibrosis to treatment poses a challenge in controlling fibrotic remodeling [9]. The potential benefits of early exercise training on cardiac fibrosis have been rarely studied in human participants and therefore warrant further investigation.

The primary aim was to evaluate whether early CR (initiated within 6 weeks after discharge for acute HF) is associated with improved long-term outcomes, while the secondary aims were to explore whether markers of fibrosis (e.g., lysyl oxidase-like 2) and baseline characteristics modify associations. We hypothesized that early post-discharge CR is associated with reduced all-cause mortality compared to standard care and that patients with higher levels of LOXL2—a biomarker indicative of cardiac fibrosis—derive greater benefit from CR participation.

## 2. Materials and Methods

### 2.1. Study Design and Participants

This was a prospective cohort-controlled study conducted at a tertiary referral medical center. From January 2019 to July 2022, we enrolled adult patients (≥20 years) who were hospitalized with acute heart failure (HF), classified as New York Heart Association (NYHA) functional classes II–IV, and who met the biomarker criteria for decompensated HF (BNP ≥ 100 pg/mL or NT-proBNP ≥ 400 pg/mL; NT-proBNP ≥ 900 pg/mL if atrial fibrillation was present). Additional inclusion criteria were as follows: (1) left ventricular ejection fraction (LVEF) ≤ 40% determined via echocardiography during hospitalization, (2) survival to hospital discharge, and (3) the completion of a structured Heart Failure Disease Management Program (HFDMP) prior to discharge. The HFDMP included a psychological assessment; education by a heart failure specialist nurse; and consultations with a dietitian, physiotherapist, and psychologist [10].

Exclusion criteria included the following: (1) expected life expectancy < 6 months due to malignancy or end-stage disease, (2) bedbound status > 3 months prior to admission, (3) musculoskeletal disorders precluding exercise testing, (4) severe cognitive impairment or inability to cooperate with assessments, (5) mechanical ventilator dependence, (6) terminal HF not amenable to rehabilitation, (7) severe primary valvular disease requiring surgery, and (8) refusal to participate by the patient or family.

All participants provided written informed consent. The study was approved by the Institutional Review Board of Chang Gung Medical Foundation (IRB No. 201801077B0) on 10 August 2018, and registered at ClinicalTrials.gov (NCT03782337).

We present this study in accordance with the STROBE guidelines for observational cohorts; the completed STROBE checklist can be found in Appendix A. Since our program is an exercise-based intervention, we offer a detailed description following the TIDieR guidelines, with completed checklists included in Appendix A. A flow diagram styled after STROBE (Figure 1) summarizes the processes of screening, exclusions (along with reasons), enrollment, and follow-up.

Before PSM, the patients in the non-CR group were older than those in the early-CR group (median age: 58.5 vs. 53.0 years, *p* = 0.022) and exhibited higher NT-proBNP levels (median: 4552.5 vs. 1275.0 pg/mL, *p* = 0.002) (Table 1). After PSM, the variables were well balanced between the two groups (Table 2).

### 2.2. Cardiac Rehabilitation Protocol

Patients were contacted by an HF nurse within 7 days of discharge to evaluate rehabilitation eligibility and willingness. Those who enrolled were scheduled for phase II outpatient CR within 6 weeks post-discharge. CR participation was defined as attending at least one supervised session and continuing a structured home-based program.

Supervised CR included several sessions of moderate-intensity aerobic training individualized based on cardiopulmonary exercise testing (CPET). Exercise prescription was based on either 10 bpm above the anaerobic threshold or 40–60% of peak VO_2_, adjusted biweekly according to the Borg scale (target 12–14). Resistance and flexibility components were incorporated as tolerated [11].

Patients also received written and illustrated guides for home-based CR. The program included 20–40 min of aerobic activity (e.g., walking and cycling) 3–5 times per week, plus light strength training, tailored to patient capacity. The HF nurse tracked adherence through biannual follow-up calls and reviewed CR records maintained by physiotherapists.

Patients not enrolled in CR received standard post-discharge care under the HFDMP without structured exercise guidance.

### 2.3. Biomarker Sampling and LOXL2 Analysis

Blood samples (≤10 mL) were obtained from patients after overnight fasting. The samples were centrifuged and stored at −70 °C until the analysis of serum LOXL2 levels. Serum LOXL2 concentrations were measured using ELISA kits (R&D Systems, Abingdon, UK) according to the manufacturer’s protocol. Absorbance was measured at 450 nm using a microplate reader, and concentrations were quantified using corresponding standard curves.

### 2.4. Outcome Measures

The primary endpoint was all-cause mortality. The secondary endpoints were (1) HF rehospitalization and (2) changes in KCCQ-12 scores at 6 and 12 months after discharge. Vital status and hospitalization data were gathered from electronic medical records maintained by a heart failure specialist nurse through the hospital’s information system.

### 2.5. Statistical Analysis

Sample size calculation was based on detecting a hazard ratio of 0.2319 for all-cause mortality with 80.3% power at α = 0.05, requiring 126 participants equally divided between CR and non-CR groups, in accordance with our previous research [5].

Demographic and clinical variables were compared using the chi-square or Fisher’s exact test for categorical variables and the Mann—Whitney U test for continuous variables. Propensity scores were calculated using logistic regression incorporating age, sex, comorbidities (diabetes, hypertension, dyslipidemia, ischemic heart disease, prior myocardial infarction, and atrial fibrillation), body mass index, estimated glomerular filtration rate, HF medication (≥3 guideline-directed therapies), and LOXL2 level.

A 1:1 nearest neighbor matching method without replacement was used. Balance between groups was assessed using the standardized mean difference (SMD), with SMD < 0.1 indicating an acceptable balance.

Clinical outcomes, including all-cause mortality, rehospitalization for HF, and improvements in KCCQ-12 scores at 6 and 12 months after propensity score matching (PSM), were compared between patients with and without CR using Pearson’s chi-square or Fisher’s exact test and the Mann—Whitney U test.

For an analysis of the associations between CR and outcomes, incidence rates per 1000 patient-years and incidence rate differences were calculated for all-cause mortality and rehospitalization for HF after PSM. Actuarial survival rates were estimated using the Kaplan—Meier method, and between-group differences were statistically examined using the log-rank test. A post hoc analysis was conducted to explore whether CR improved the prognosis related to cardiac fibrosis. Patients were categorized into three groups according to LOXL2 levels: 0–100 pg/mL, >100 to ≤200 pg/mL, and >200 pg/mL. We reference benchmark thresholds from previous research, such as the findings in a Nature Communications report [12] that identified a range of 90 to 100 pg/mL in discriminating HFrEF from control subjects. Serum LOXL2 levels showed a correlation with those of TIMP-1 and ST-2, both of which are markers associated with cardiac remodeling and fibrosis. Serum LOXL2 levels greater than 200 pg/mL may indicate cases of more severe cardiac fibrosis. Additionally, incidence rates per 1000 patient-years and incidence rate differences for all-cause mortality were calculated to analyze the association between CR and outcomes and stratified by LOXL2 levels at admission.

All statistical analyses were performed using SPSS version 25.0 (IBM Corp., Armonk, NY, USA), with two-tailed *p* < 0.05 considered statistically significant.

## 3. Results

### 3.1. Patient Characteristics

A total of 480 patients with acute HF participated in the HFDMP from January 2019 to July 2022. Among them, 162 patients met both the inclusion and exclusion criteria and agreed to participate in this study. Overall, the early-CR group comprised 34 patients who participated in the phase II exercise training program and continued to undergo home-based CR. In the CR group, 34 patients participated in center-based CR, with a median of 7.0 rehabilitation exercise sessions (IQR: 1.00–27.25). The reasons for not participating in CR include personal preference, transportation issues, caregiver constraints, and medical restrictions. PSM resulted in 33 patients per group (Figure 1).

### 3.2. Clinical Outcomes

After PSM, the patients were evenly divided into the early-CR and non-CR groups (Table 2). The all-cause mortality rate was significantly lower in the early-CR group (0% vs. 21.2%, *p* = 0.011) (Table 3A). No significant differences in the secondary endpoints, which included rehospitalization for HF (*p* = 0.269), improvement in KCCQ-12 scores at 6 months (*p* = 0.387), and improvement in KCCQ-12 scores at 12 months (*p* = 0.934), were identified between the groups (Table 3A).

During a median follow-up period of 2.85 years (IQR: 2.33–3.83), patients who underwent CR showed a lower incidence rate of all-cause mortality (0 vs. 87.16 events per 1000 patient-years; rate difference: −0.087 [95% CI: −0.143 to −0.031]; *p* = 0.002) (Table 3B). However, the beneficial effect of CR on rehospitalization for HF did not reach statistical significance (82.08 vs. 160.90 per 1000 patient-years; rate difference: −0.079 [95% CI: −0.188 to −0.030]; *p* = 0.156) (Table 3B). Patient survival rates according to participation in CR were examined using Kaplan—Meier curves, and p-values for differences between the early-CR and non-CR groups were calculated using the log-rank test. The log-rank test indicated that CR was associated with a significantly reduced incidence of all-cause mortality (*p* = 0.004) (Figure 2A); conversely, CR was not associated with a lower incidence of rehospitalization for HF (*p* = 0.264) (Figure 2B).

Patients with higher LOXL2 levels (>200 pg/mL) at admission benefited from CR (0 vs. 172.3 events per 1000 patient-years; rate difference: −0.172 [95% CI: −0.299 to −0.046]; *p* = 0.008) (Table 4). However, the benefits of CR in patients with lower LOXL2 levels (0–100 pg/mL and >100 to ≤200 pg/mL) were not statistically significant (*p* = 0.357 and *p* = 0.068, respectively) (Table 4).

During the CR period, no exercise-related deaths or major adverse events (e.g., arrhythmia and falls) occurred. Furthermore, no significant cardiovascular events associated with the home-based exercise program were reported. Additionally, exercise did not lead to the worsening of HF symptoms that would require hospitalization.

## 4. Discussion

This prospective cohort study revealed that an early exercise-based CR program initiated within 6 weeks of acute HF admission was associated with lower all-cause mortality over a nearly 3-year follow-up period compared with no CR; HF rehospitalization and quality of life status (KCCQ-12 scores) were not significantly different between groups. Notably, patients with more severe cardiac fibrosis derived significant benefits from the CR program. To the best of our knowledge, this is the first prospective study to demonstrate that an early exercise-based CR program with a continued home-based CR program can reduce long-term all-cause mortality in patients discharged after an acute HF episode.

More than half of patients readmitted for HF within 30 days do not consult a physician or nursing specialist, and those who do not receive outpatient follow-up within 7 days have the highest 30-day readmission rate [13]. The HFDMP, led by HF nursing specialists, has been shown to improve medication adherence and risk factor modification, effectively reducing readmission rates [14]. In our study, both groups participated in an HFDMP led by an HF nursing specialist, which contributed to reduced readmission rates in both groups. The other explanations for the dissociation between survival and HF rehospitalization are limited power for cause-specific events, selective effects of post-discharge optimization (including GDMT titration) on mortality, and competing risk dynamics. Null KCCQ-12 differences may relate to measurement timing, regression to the mean, and ceiling/floor effects; responder analyses using the ≥5-point threshold help contextualize potential subclinical gains. However, only the early-CR group participated in the exercise- and center-based phase II CR program. This study demonstrates that integrating an early exercise program into the HFDMP, initiated within 6 weeks of discharge, could further significantly reduce all-cause mortality over a nearly 3-year follow-up period.

The early initiation of exercise training to prevent further physical deconditioning is considered beneficial for patients with HF. In the REHAB-HF trial, exercise was introduced during hospitalization; however, the study lacked power to assess clinical outcomes, and mortality appeared higher in patients with HFrEF [6]. A systematic review and meta-analysis of 13 RCTs found that early exercise-based CR improved exercise capacity (6-minute walk distance, SPPB), quality of life (Minnesota Living with Heart Failure Questionnaire, KCCQ), and daily activities and reduced all-cause readmissions, but did not significantly affect LVEF, HF readmissions, or mortality [15]. Clinical guidelines recommend early ambulation and exercise after acute cardiovascular events, including acute HF [16,17]. A nationwide Japanese registry of 10,473 patients reported that early exercise was associated with a lower composite outcome (291 vs. 327 events per 1000 patient-years; rate ratio 0.890, 95% CI: 0.830–0.954; *p* = 0.001), although cardiovascular mortality was unchanged [18]. Similarly, most studies with short follow-up (<6 months) have not shown mortality benefits [19,20]. In contrast, a large U.S. retrospective cohort (40,364 patients) demonstrated that exercise-based CR was linked to a 42% reduction in all-cause mortality at 2 years (odds ratio: 0.58, 95% CI: 0.54–0.62) [21]. Another prospective cohort with PSM (626 patients) also showed lower mortality (HR: 0.53, 95% CI: 0.30–0.95) over 2 years, though survival benefits were not evident beyond 6 months [22]. Our prospective study, with longer follow-up and a home-based CR component, further supports the role of early exercise-based CR after acute HF discharge in significantly reducing all-cause mortality.

Patients with HF accompanied by cardiac fibrosis experience considerably worse outcomes [7], and few studies have demonstrated substantial benefits from either pharmacological treatments or non-pharmacological management in human participants [8,9]. There remains an unmet need to target maladaptive left ventricular tissue remodeling and fibrosis in order to improve the systolic and diastolic function of the myocardium [23]. Cardiac fibrosis not only disrupts the electromechanical coordination of cardiomyocytes, reducing systolic function, but also increases ventricular stiffness, impairing diastolic function [24]. LOXL2 is upregulated in the interstitial tissue of failing hearts in both mice and humans. An elevated LOXL2 expression promotes transforming growth factor (TGF)-β2 production, inducing myofibroblast formation and migration, causing enhanced collagen deposition and crosslinking in hypertrophic regions of the stressed heart [25]. These pathological processes contribute to the development of cardiac fibrosis, ultimately resulting in myocardial dysfunction. Targeting LOXL2 in HF therapy is supported by findings from cell-based assays, pharmacological studies, mouse genetic models, and clinical evidence in patients with HF [26]. To the best of our knowledge, this is the first study to demonstrate that exercise-based CR provides significant benefits to patients with HF accompanied by more severe cardiac fibrosis.

LOXL2 measurement is currently available predominantly as research-use-only ELISA platforms rather than cleared IVD (In Vitro Diagnostic medical device) assays, with per-kit costs in the hundreds of USD and no established clinical reimbursement. While circulating and myocardial LOXL2 associate with fibrosis and HF biology, therapeutic targeting of LOXL2 has not translated to improved clinical outcomes in fibrotic diseases to date [26]. Accordingly, we view LOXL2 as a candidate stratification biomarker for prospective trials rather than a test ready for routine CR triage.

Exercise enhances cardiopulmonary function, thereby improving patient survival. Potential mechanisms include an increased central cardiac output, enhanced peripheral oxygen consumption, better musculoskeletal function [27], and improved vascular structure and function [28]. Exercise may also provide cardioprotective effects against cardiac fibrosis. The mechanism by which exercise mitigates cardiac fibrosis may involve promoting the secretion of cardioprotective exerkines, inhibiting systemic overactivation of the sympathetic nervous system and the renin—angiotensin system, attenuating oxidative stress and inflammatory responses, and modulating metabolism and noncoding RNA expression [29]. In various animal studies, aerobic exercise has been demonstrated to provide potential benefits in mitigating cardiac fibrosis. Experimental studies have shown that exercise stimulates the release of plasma exosomes from endothelial progenitor cells, which are enriched with microRNAs, such as miR-126 [30]. These exosomes exhibit cardioprotective properties by activating cardiac fibroblasts, enhancing the expression of endothelial cell-specific markers, and suppressing proteins associated with fibrosis [31]. The exercise-induced release of exosomes, which mitigates cardiac fibrosis, provides a novel mechanism underlying the therapeutic benefits of CR.

### 4.1. Implications and Perspectives for Clinical Practice

The findings of this study indicate that the early initiation of CR within six weeks of hospital discharge is associated with a significant reduction in all-cause mortality among patients with acute HF, particularly those with coexisting cardiac fibrosis. These results underscore the clinical importance of timely referral to structured exercise-based rehabilitation programs as a critical component of post-discharge management. The integration of early CR into standard care pathways may enhance clinical outcomes and alleviate healthcare resource utilization, especially in patients with fibrotic cardiac remodeling.

From a clinical practice standpoint, the core elements enabling early cardiac rehabilitation can be embedded within clinical networks to advance the Quadruple Aim (reducing costs; improving population health, patient experience, and team well-being) [32]. Systematic reviews indicate that clinical networks improve effectiveness and efficiency, while multidisciplinary disease management models streamline pathways and reduce adverse events—features directly relevant to post-discharge HF care [33]. Concurrently, patient-centered care frameworks in acute settings are linked with better experiences and, in some contexts, improved outcomes, supporting our emphasis on individualized goal-setting, shared decision-making, and adherence support.

### 4.2. Limitations

This study has some limitations. First, while our study provides real-world data, it is a small cohort study conducted at a single medical center with an open-label design. Our single-center design and local care pathways may limit generalizability; we propose multicenter implementation within managed clinical networks to enhance external validity. Second, the decision to participate in CR was based on patients’ personal choice, which might have led to a greater likelihood of those with better overall conditions opting for CR. The potential baseline imbalances in comorbidities, medications, or functional status among HF patients could confound the observed associations. Despite the use of a PSM analysis to make the evaluation more objective, this selection bias may still have influenced the outcomes. Third, the outbreak of the COVID-19 pandemic occurred during our enrolment period, and home-based CR for the CR group was implemented. However, adherence to home-based CR could not be fully monitored [34]. Previous studies have shown that telemedicine and wearable monitoring can enhance patients’ cardiorespiratory fitness and cardiac function by improving adherence, which was not utilized in this study [35]. Although we expended efforts to track patients using heart rate variability and video consultations, this was not as precise as center-based CR. Nevertheless, this experience highlights that home-based CR could be a potential area for future development. Fourth, the effectiveness of CR may follow a dose—response relationship, which was not evaluated in this study. However, in our previous publication, we observed a dose—response relationship [5]. Fifth, comprehensive follow-up assessments of patients’ CPET, which could serve as an important prognostic indicator, were not performed. The lack of serial CPET follow-up may be attributed to resource limitations, restrictions during the pandemic, and patient willingness to participate. Future studies should incorporate serial CPET to measure the mechanistic benefits, as increases in peak VO_2_ are strongly linked to better outcomes in HF patients. We also did not collect a validated pre-admission physical-activity instrument. Because prior activity may influence capacity to engage in CR and outcomes, residual confounding is possible. Sixth, our enrollment coincided with the dissemination of the 2021 ESC heart-failure guidelines and the 2023 update, resulting in lower adoption rates of ARNI and SGLT2i compared to current practices. Given that GDMT impacts exercise tolerance and cardiac remodeling, the effectiveness of early CR may differ in fully optimized, contemporary cohorts. Future research should investigate CR within standardized, rapidly implemented GDMT frameworks. We did not conduct formal sensitivity analyses for unmeasured confounding or for missing data. This limits the robustness of our findings and highlights their exploratory nature. Lastly, our study did not analyze psychological factors, including depression and cognitive function, which are critical components of a multidisciplinary CR program [10]. Future large-scale studies incorporating these factors should be conducted for a more comprehensive evaluation.

### 4.3. Future Directions

We propose a multicenter, assessor-blinded RCT enrolling adults hospitalized for acute HF who are eligible for early CR. Randomization (1:1) to CR + usual care vs. usual care will be stratified by baseline LOXL2 and myocardial fibrosis burden (LGE extent and/or ECV by T1 mapping). Co-primary endpoints include (1) 12-month HF hospitalization or all-cause death and (2) change in peak VO_2_ at 3–6 months. Secondary endpoints include changes in ECV, LGE volume, quality of life, and fibrosis biomarkers (LOXL2, galectin-3, sST2).

## 5. Conclusions

This study found that the early initiation of exercise-based CR within six weeks of hospital discharge was associated with improved long-term survival in patients with acute HF. Implementation within multidisciplinary disease management networks and with patient-centered processes may enhance adherence and outcomes. The survival benefit was particularly evident in those with elevated LOXL2 levels, suggesting that LOXL2 may serve as a clinically relevant biomarker to identify patients with cardiac fibrosis who are most likely to benefit from structured CR interventions. LOXL2 should be regarded as a hypothesis-generating biomarker, not ready for routine clinical use. Programs should prioritize the prompt referral of CR and coordinate with HFDMP. Multicenter randomized trials in the current guidelines’ era are necessary to confirm effectiveness across diverse care settings.

## Figures and Tables

**Figure 1 biomedicines-13-02228-f001:**
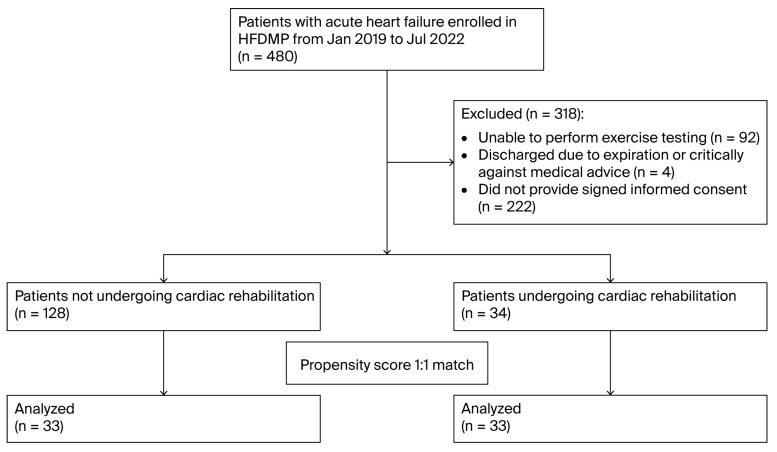
Flowchart of the study. HFDMP, Heart Failure Disease Management Program.

**Figure 2 biomedicines-13-02228-f002:**
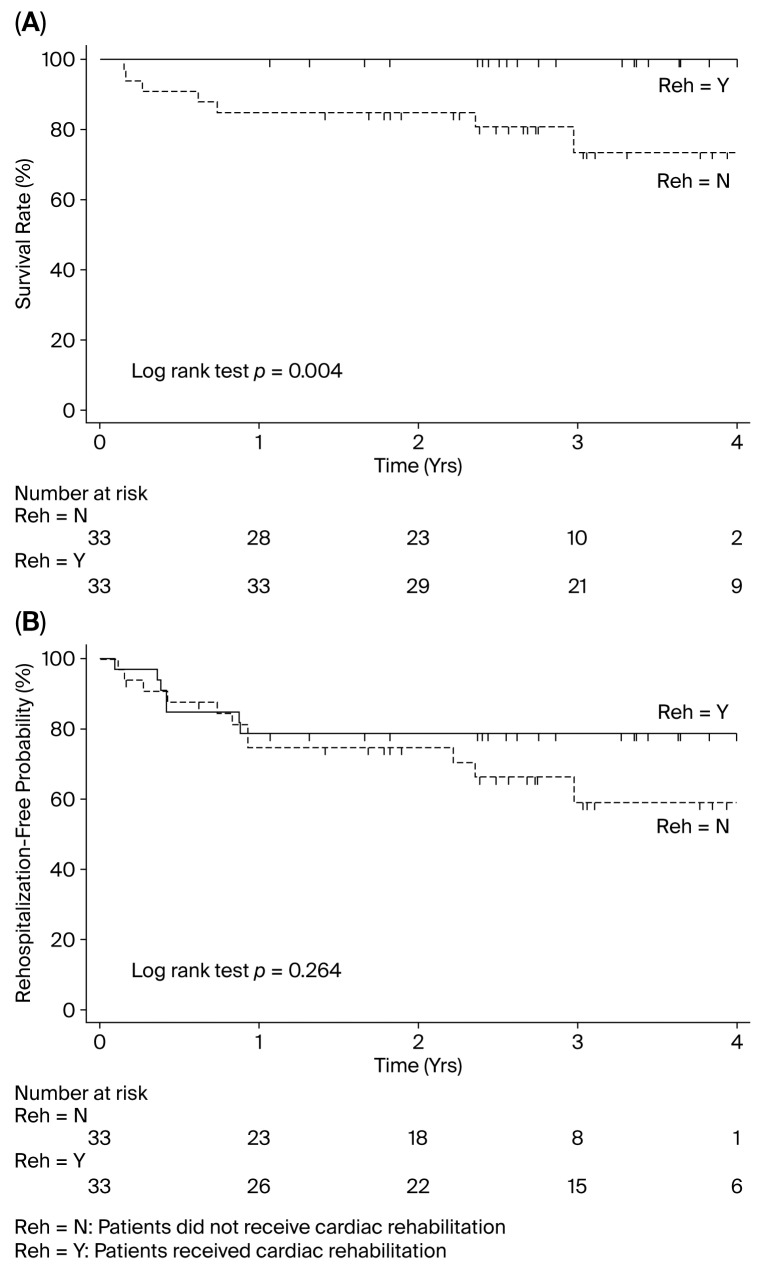
Cumulative incidence curves for (**A**) all-cause mortality and (**B**) rehospitalization for HF in a propensity score-matched cohort. HF, heart failure. Reh, rehabilitation.

**Table 1 biomedicines-13-02228-t001:** Demographic data of the study population.

Demographic Characteristics	Early CR *n* = 34 *n* (% *N*) or Median (Range)	Non-CR *n* = 128 *n* (% *N*) or Median (Range)	*p*-Value
Age, median (range)	53.00 (42.50–61.50)	58.50 (48.00–67.00)	0.022
Sex, male	27 (79.4%)	104 (81.3%)	0.809
Ischemic CM	25 (73.5%)	86 (67.2%)	0.479
Diabetes mellitus	11 (32.4%)	51 (39.8%)	0.424
Hypertension	20 (58.8%)	85 (66.4%)	0.411
Hyperlipidemia	8 (23.5%)	45 (35.2%)	0.199
Previous MI	2 (5.9%)	6 (4.7%)	0.674
Atrial fibrillation	3 (8.8%)	8 (6.3%)	0.701
Body mass index (kg/m^2^), median (range)	24.69 (22.69–29.75)	25.88 (22.54–30.00)	0.789
Medication			
ACEI or ARB	23 (67.6%)	85 (66.4%)	0.891
ARNI	7 (20.6%)	33 (25.8%)	0.533
Beta-blocker	29 (85.3%)	111 (86.7%)	0.784
MRA	24 (70.6%)	76 (59.4%)	0.232
SGLT2i	4 (11.8%)	11 (8.6%)	0.521
Digoxin	3 (8.8%)	13 (10.2%)	1.000
Diuretics	20 (58.8%)	92 (71.9%)	0.143
Mean BP (mmHg)	96.50 (82.25–105.25)	97.00 (87.00–110.00)	0.289
Heart rate (bpm)	93.50 (76.00–104.25)	91.00 (76.00–102.00)	0.824
LVEF (%) at baseline	27 (20–34)	27 (22–33)	0.784
Estimated glomerular filtration rate	74.45 (57.58–93.45)	66.18 (43.58–87.70)	0.059
Creatinine (mg/dL)	1.07 (0.89–1.30)	1.22 (0.89–1.69)	0.139
Hemoglobin (g/L)	13.35 (12.15–15.45)	13.65 (11.93–15.20)	0.673
NT-proBNP (pg/mL)	1275.00 (659.43–5157.50)	4552.50 (1478.75–8475.00)	0.002
T3	69.15 (60.18–79.43)	69.05 (55.75–82.08)	0.681
fT4	1.07 (0.94–1.20)	1.06 (0.97–1.17)	0.930
TSH	1.95 (1.29–3.34)	2.01 (1.20–3.26)	0.852
Ferritin	151.50 (67.30–308.08)	202.10 (73.85–414.55)	0.128
Peak VO_2_	17.30 (13.50–21.50)	16.45 (12.70–18.80)	0.252
LOXL2 (pg/mL)	133.05 (91.90–220.83)	110.15 (87.05–151.18)	0.141

CR, cardiac rehabilitation; CM, cardiomyopathy; MI, myocardial infarction; ACEI, angiotensin-converting enzyme inhibitor; ARB, angiotensin receptor blocker; ARNI, angiotensin receptor—neprilysin inhibitor; MRA, mineralocorticoid receptor antagonist; SGLT2i, sodium—glucose cotransporter-2 inhibitor; BP, blood pressure; NT-proBNP, N-terminal pro-B-type natriuretic peptide; TSH, thyroid-stimulating hormone; LOXL2, lysyl oxidase-like 2.

**Table 2 biomedicines-13-02228-t002:** Demographic data of the study population after 1:1 propensity score matching.

Demographic Characteristics	Early CR (*n* = 33) *n* (% *N*) or Mean (SD)	Non-CR (*n* = 33) *n* (% *N*) or Mean (SD)	Standardized Mean Difference
Age, median (range)	52.67 (11.35)	51.82 (13.93)	0.072
Sex, male	26 (78.8%)	27 (81.8%)	0.076
Ischemic CM	24.0 (72.7%)	24.0 (72.7%)	<0.001
Diabetes mellitus	11.0 (33.3%)	11.0 (33.3%)	<0.001
Hypertension	20.0 (60.6%)	18.0 (54.5%)	0.123
Hyperlipidemia	8.0 (24.2%)	7.0 (21.2%)	0.072
Previous MI	2.0 (6.1%)	1.0 (3.0%)	0.146
Atrial fibrillation	2.0 (6.1%)	2.0 (6.1%)	<0.001
Body mass index (kg/m^2^), mean (SD)	26.50 (5.73)	26.27 (6.29)	0.039
GDMT (≥3)	21.0 (63.6%)	19.0 (57.6%)	0.124
Estimated glomerular filtration rate, mean (SD)	72.87 (25.21)	75.22 (26.17)	0.091
LOXL2 (pg/mL), mean (SD)	152.09 (77.64)	137.12 (70.07)	0.202

CR, cardiac rehabilitation; CM, cardiomyopathy; MI, myocardial infarction; SD, standard deviation; LOXL2, lysyl oxidase-like 2.

**Table 3 biomedicines-13-02228-t003:** Clinical outcomes in patients with or without rehabilitation (propensity score-matched controls). (**A**) Primary and secondary endpoints. (**B**) Event and incidence rate differences.

(A)				
	Early CR (*n* = 33)		Non-CR (*n* = 33)	*p*
All-cause mortality	0 (0)		7 (21.2)	0.011
Rehospitalization for HF	7 (21.2)		11 (33.3)	0.269
Improvement in KCCQ-12 scores at 6 months	25.52 (5.73–40.94)		32.12 (10.68–46.59)	0.387
Improvement in KCCQ-12 scores at 12 months	27.09 (8.86–47.61)		31.51 (9.16–45.44)	0.934
(**B**)				
**Outcomes**	**Event Rate** **(1000 Patient-Years)**		**Incidence Rate Difference (95% CI)**	***p*-Value**
	**Early CR**	**Non-CR**		
All-cause mortality	0	87.16	−0.087 (−0.143 to −0.031)	0.002
Rehospitalization for HF	82.08	160.90	−0.079 (−0.188 to −0.030)	0.156

CR, cardiac rehabilitation; HF, heart failure; KCCQ-12, 12-item Kansas City Cardiomyopathy Questionnaire; CI, confidence interval; HF, heart failure.

**Table 4 biomedicines-13-02228-t004:** Effects of CR on all-cause mortality according to LOXL2 level at admission.

Subgroups (*n*)	Even Rate (1000 Patient-Years)		Incidence Rate Difference	*p*-Value
	Early CR	Non-CR		
LOXL2 level: 0–100 pg/mL (*n* = 61)	0	31.29	−0.031 (−0.098 to 0.035)	0.357
LOXL2 level: >100, ≤200 pg/mL (*n* = 70)	0	80.74	−0.081 (−0.168 to 0.006)	0.068
LOXL2 level: >200 pg/mL (*n* = 31)	0	172.3	−0.172 (−0.299 to −0.046)	0.008

CR, cardiac rehabilitation; LOXL2, lysyl oxidase-like 2.

## Data Availability

The data underlying this article will be shared on reasonable request to the corresponding author.

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
