# Peer review of "Lysyl Oxidase-like 2-Guided Benefits of Early Cardiac Rehabilitation in Acute Heart Failure: A Prospective Cohort Study in Taiwan"

_biomedicines, 2025, doi:10.3390/biomedicines13092228_

Round 1
Reviewer 1 Report
Comments and Suggestions for Authors
Dear Author,
Below are my suggestions on the paper
I recommend that the authors should discuss deep into the selection bias and the implications of a single-center design, particularly regarding how these factors might influence the generalizability of the findings. The authors can suggest some strategies for future research to tackle these issues, such as conducting multicenter randomized controlled trials (RCTs).
Authors need to clarify the reasoning behind the LOXL2 cutoff levels used in the subgroup analysis. It is required to cite previous studies or providing data on how these thresholds relate to fibrosis severity or clinical outcomes. This would strengthen the section and understanding for readers.
More information on how adherence to home-based cardiac rehabilitation (CR) was measured would be beneficial. Additionally, discussing how varying levels of adherence might impact outcomes is crucial. If available, including data on adherence rates or patterns would add value.
I recommend considering a secondary analysis to see if there's a correlation between the number of CR sessions or the intensity of exercise and the outcomes. Even if this is exploratory, it could provide valuable insights into the best CR protocols.
It would be important to address why follow-up cardiopulmonary exercise testing (CPET) or other functional assessments were not conducted. Discussing how including these assessments in future studies could bolster the evidence for the mechanistic benefits of CR would be advantageous.
Finally, I suggest proposing specific designs for randomized controlled trials to validate these findings. This could include larger sample sizes, multicenter settings, and the incorporation of imaging or additional biomarkers to confirm the role of LOXL2.
Recommendation:
Accept with Revisions: This study presents novel and clinically significant findings regarding the advantages of early CR in acute heart failure, especially for patients with elevated LOXL2 levels. The methodology is robust, and the results are persuasive. However, the limitations—such as the small sample size, selection bias, single-center design, and incomplete adherence monitoring—call for minor revisions to enhance the discussion and clarify the findings. Addressing the recommendations above will improve the paper’s scientific quality and impact, making it a strong candidate for publication in Biomedicines.
Reviewer 2 Report
Comments and Suggestions for Authors
Dear authors,
Thank you for the opportunity. Under the improvements required:
- Title: missing the setting (e.g. "German") and isn't recommended the acronim use in this part;
- Abstract: I suggest a clinical view discussion of this section as for the discussion sections in the text;
- Keywords: see the comment for the title for type of study and setting (see the title suggest for acronim use);
- Introduction: missing full and clear epidemiological data in international and national perspective at the beginning of the section. Missing a relevant element in clinical practice view (see the comment for the discussion). The objectives aren't full clear (line 79-84): I suggest classical formula as "the primary aim was... while the secondary was/were...";
- Method: missing reporting tool (mandatary for scientific community) that could help the authors to develop a international framework for this relevant section. This version is poor in relevant elements as for esemple the recruiting. Is necessary to add the reference of appropriate tool used and check list in supplementary file (e.g. Equator Network);
- Discussion: complete the section, mainly in clinical practice view. I suggest to extend the section "Clinical Implication" and combine the sections and possible change of title in "Implication and perspective for Clinical Parctice". I would like to congratulate you for your interesting contribution to the topic of early cardiac rehabilitation in patients with acute heart failure, which in my view addresses an area of great clinical and scientific relevance. However, I believe that including some complementary perspectives could further enrich the opening section and help give the manuscript an international scope, thereby increasing its chances of consideration in leading editorial venues. In particular, it may be useful to integrate references exploring the impact of clinical networks and organizational models in improving outcomes and the sustainability of rehabilitation programs, such as "Achieving quadruple aim goals through clinical networks", "Practice of Multidisciplinary Collaborative Chain Management Model", and "Development of patient-centred care in acute hospital settings". These additions could harmoniously complete the discussion and strengthen the overall message of your study, in line with current trends in the international literature;
- Limitations: missing possible generalizability of data finding for population and tools adopted;
- Conclusion: according to the previous suggestions, required update in merit; References: according to previous suggestions, consider to update the references over 10 years not adopted for method or with high impact of evidence.
- Editing: I'm not sure that the references are according to journal template, please consider for clarify the number and title of subsection (e.g. 2.1 Study design and participants...), the flow study isn't clear, the tables required editing improvements for clarify and not all acronim used is indicated in the legend of single table (please full check full manuscript for this element), please adopt "gender" and not "sex" in the tables and in the text too;
- This version merit improvements that I required foundamental for possible international considerationin method and clinical practive view.
Native review recommended
Reviewer 3 Report
Comments and Suggestions for Authors
This manuscript addresses an important and timely clinical question – the role of early cardiac rehabilitation (CR) in patients after acute heart failure (HF). The study is prospective, methodologically sound, and includes propensity score matching, which strengthens the robustness of the analysis. The originality of the work is underlined by the use of LOXL2 as a biomarker of fibrosis to stratify patient benefit. The results are clearly presented and clinically meaningful.
Study limitations
The authors emphasize the impact of the COVID-19 pandemic as a limitation. However, I believe the more relevant issue is the timing of the study, which coincided with the publication of updated ESC HF guidelines (2021 and 2023). This explains the relatively low proportion of patients treated with ARNI and SGLT2 inhibitors. The authors should explicitly mention that this factor influenced treatment patterns and may limit the generalizability of results.
Impact of pharmacotherapy
As shown in Tables 1 and 2, only about 20% of patients received ARNI and approximately 10% SGLT2i. This is considerably lower than expected according to current standards of care. Since background pharmacotherapy directly affects exercise tolerance and response to rehabilitation, it should be clearly stated that the observed effectiveness of CR may differ in patients treated according to the latest guidelines.
Baseline physical activity
The study does not provide information regarding patients’ physical activity prior to hospitalization. This is an important limitation, since prior activity level could significantly affect the capacity to engage in CR and influence outcomes. Even a simple stratification (sedentary vs. moderately active) would add value to the analysis.
Tables and data consistency
Numerical consistency: There are some discrepancies between the patient numbers reported in text and in tables before and after PSM – this should be carefully reviewed and unified.
Data presentation: Some variables are expressed as median (IQR), while others as mean ± SD. Please consider unifying the reporting format or provide a rationale for the differences.
Table 3A/3B: The survival benefit is clear, yet there were no significant differences in HF rehospitalization or in KCCQ-12 scores. This important finding should be discussed more explicitly.
Editorial remarks
The discussion could be slightly condensed by removing repetitions, which would improve clarity.
The conclusion would benefit from a more practice-oriented statement, emphasizing the need for prompt referral to CR and the importance of future randomized trials in the context of contemporary pharmacotherapy.
This is a valuable and original manuscript, presenting novel insights into early cardiac rehabilitation and fibrosis-guided patient selection. I recommend acceptance after maior revision.
Reviewer 4 Report
Comments and Suggestions for Authors
The Authors investigated whether early initiation of CR within 6 weeks of discharge improves long-term outcomes in patients hospitalized with acute HF and whether baseline LOXL2 levels modify this effect. In a prospective cohort of 162 patients enrolled in a structured disease management program, 34 underwent early CR and were compared with non-CR patients using propensity score matching. Over a median follow-up of nearly 3 years, early CR was associated with improved survival, with the most pronounced benefit observed among patients with elevated LOXL2 levels, suggesting a potential role for this biomarker in identifying patients most likely to benefit from CR.
Major concerns:
-The Authors need to clarify how CR adherence and intensity were monitored and whether variations could have influenced the outcomes.
-They should report on potential baseline imbalances in comorbidities, medications, or functional status that could confound the observed associations despite propensity score matching.
-The Authors should provide additional details on the characteristics of patients who did not participate in CR, as selection bias may have influenced the findings.
-Data should be provided on secondary outcomes, particularly HF re-hospitalization and KCCQ-12 changes, as these are clinically important endpoints.
-Proper randomized controlled trials should be performed to confirm the survival benefit of CR and validate the role of LOXL2 as a stratification biomarker.
-The following pertinent reports should be discussed: doi: 10.3390/jcdd12010008, doi: 10.1093/eurjpc/zwae324, doi: 10.1093/eurjpc/zwae363, doi: 10.1124/jpet.122.001149, doi: 10.3390/jcdd11090255, doi: 10.1186/s12933-024-02425-6.
-The Authors need to at least discuss whether LOXL2 testing is feasible, cost-effective, and ready for clinical application before proposing it as a guide for CR strategies.
Round 2
Reviewer 2 Report
Comments and Suggestions for Authors
Dear Editor,
very god job. In this second version ready for publication. Congratulation because your work is very interesting for the scientific community.
Best
Author Response
Thank you for your feedback.
Reviewer 3 Report
Comments and Suggestions for Authors
the authors are making corrections and clarifications, I believe the manuscript can now be considered for publication
Author Response
Thank you for your feedback.
Reviewer 4 Report
Comments and Suggestions for Authors
The Authors were only partially responsive.
Major issue remain.
Author Response
Thank you for your kind reviews.